# Exploration of Changes in Low-Income Latino Families’ Beliefs about Obesity, Nutrition, and Physical Activity: A Qualitative Post-Intervention Study

**DOI:** 10.3390/bs12030073

**Published:** 2022-03-09

**Authors:** Rochelle Cason-Wilkerson, Shauna Goldberg Scott, Karen Albright, Matthew Haemer

**Affiliations:** 1Section of Nutrition, Department of Pediatrics, School of Medicine, University of Colorado Anschutz Medical Campus, Aurora, CO 80045, USA; matthew.haemer@cuanschutz.edu; 2Institute for Health Research, Kaiser Permanente Aurora, Aurora, CO 80014, USA; shauna.scott@kp.org; 3Department of Internal Medicine, School of Medicine, University of Colorado Anschutz Medical Campus, Aurora, CO 80045, USA; karen.albright@cuanschutz.edu

**Keywords:** childhood obesity, family-inclusive treatment, change in beliefs, low-income, Latino

## Abstract

Objective: To investigate changes in beliefs around obesity, nutrition, and physical activity among low-income majority Latino families who participated in a community-based family-inclusive obesity intervention. Methods: Six focus groups were conducted with a predominately Latino low-income population, who completed the Healthy Living Program (HeLP). Two groups were conducted in English and four groups were conducted in Spanish, and were recorded, translated, transcribed, and analyzed for thematic content. Two coders independently coded transcripts then reflexive team analysis with three members was used to reach consensus. Results: Thirty-seven caregivers representing thirty-three families participated in focus groups. A number of themes emerged around changes in beliefs about obesity, nutrition, and physical activity (PA) as a result of the HeLP curriculum. Regarding obesity, the themes that emerged focused on the acceptability of children being overweight and the importance of addressing weight at an early age. Changes in beliefs regarding nutrition emerged, noting changes in the use of food as a reward, the multiple benefits of a healthy diet, and for some participants change in their beliefs around the adaptability of traditional foods and habits. Regarding physical activity, themes emerged around the difficulty of engaging in PA due to unsafe conditions and finding creative indoor and outdoor activities with whole family participation and becoming aware of the benefits of PA. Conclusions: Parental changes in beliefs about obesity, nutrition, and physical activity as a result of a family-inclusive weight management program in a population of low-income predominately Latino families can aid and inform the development of future weight management programs for this population.

## 1. Introduction

Childhood obesity has been widely recognized as a major public health problem for more than two decades, and progress toward reversing this epidemic has been unacceptably slow, especially for children from low-income and minority groups. The prevalence of obesity remains higher among Hispanic/Latino children of all ages (20.9%) than among non-Hispanic whites (16.9%) in the U.S. [1,2]. Latino children are also disproportionately affected by co-morbidities of obesity, such as type II diabetes, increase in cardiovascular risk factors, hypertension, and fatty liver disease [3]. Low family income is associated with a higher prevalence of obesity for Hispanic children, who are more than twice as likely to live in poverty than non-Hispanic white children in the U.S. [4]. For example, children who experience poverty, whether at birth, intermittently, or chronically in early life, experience obesity at more than double the rate of those children who do not experience poverty during childhood [5]. Though Latino children face marked disparities related to obesity and are the fastest-growing minority group in the U.S., few studies have identified effective interventions to help to combat childhood obesity in this population, especially among parents who are primarily Spanish speakers [6].

Parental acculturation as well as parental behaviors play a large role in obesogenic lifestyle habits that may lead to weight gain; because this parental behavior plays a significant role in the risk for childhood obesity, parental engagement is critical to effective interventions [7]. Previous qualitative research has documented some perceived barriers affecting Latino parents’ capacity to implement healthy lifestyle changes for or with their children [8,9]. Having a unique family-inclusive intervention allowed for not only the child and parent, but siblings and other caretakers to all be involved, allowing for maximum parent and caregiver participation as well as the modeling of those behaviors.

For many Latino families, beliefs about food, feeding practices, and weight status undergo a transition with longer residence in the United States. National Health Interview Survey data show that Latino immigrant adults living in the U.S. for more than 15 years have a four-fold increased risk of obesity compared to shorter-term residents [9]. Studies of Latino parents have identified potential contributors to childhood obesity relevant to Latino culture, including (1) traditional foods and eating habits, (2) the influence that extended family members have on child feeding, and (3) beliefs that a heavier child is a healthier child [10]. Regarding traditional foods and eating habits, Latino parents report a prevalent desire to have children “clean their plate”, which has a persistent influence on child eating habits. In relationship to the influence of extended family members, grandparents especially may impose beliefs about child feeding that hold great influence on parents. Traditional beliefs regarding a heavier child being healthier have also led some parents to dismiss concerns about overweight status and to report their concern that healthy weight and even overweight children are underweight [9]. Additionally, while some Latino parents continue cooking traditional meals, there is also an increased exposure to the American food landscape and eating habits that occurs with acculturation. This change in eating practices may also be compounded by a more hectic pace of life in the U.S. relative to their country of origin [9].

Behavioral theories provide a structure that can help clarify how beliefs influence behaviors. The Theory of Planned Behavior holds that the perceived value of change, normative beliefs about change, and self-efficacy to change are important in initiating and sustaining health behavior changes, such as those targeted in families with unhealthy weight children [10,11]. Obesity treatment interventions have rarely been evaluated qualitatively, in a way that helps to understand how and why transitions in these beliefs might occur, especially among demographic groups facing obesity-related health disparities. This study is unique in using the Theory of Planned Behavior constructs to examine the experience of low-income, predominantly Latino parents who, along with their children, participated in a family-inclusive intervention to treat childhood obesity. In using this theory to construct a moderator guide, we believe, after analyzing the parent’s responses, that the questions allowed for the reflection of parents’ normative beliefs and how those initial normative beliefs changed post-intervention. This family-inclusive intervention is distinctive in that it included the obese or overweight child, but additionally invited the siblings of the child, parents, as well as other caregivers, such as grandparents and aunts. While previous qualitative studies have examined the perspectives of school-aged children and adolescents who participated in obesity treatment in a parent–child dyad, few have focused on the entire Latino family [12,13,14,15,16,17]. Short-term results demonstrating improvement in child BMI have been previously published and a large-scale randomized-controlled trial is underway to examine the long-term effects of the program [18]. (ClinicalTrials.gov Identifier: NCT05041855).

The objective of this qualitative study is to examine the post-intervention reflections of parents regarding the Healthy Living Program. The questions used in the focus groups were influenced by the Theory of Planned Behavior (see Appendix A for the complete moderator guide). This study aims to contribute to the literature by documenting the experience of Latino families in weight management, exploring if beliefs relevant to weight and lifestyle habits changed during treatment and which components of the whole family intervention may have helped them to evolve. In lieu of any a priori hypotheses, we seek to identify beliefs whose evolution seemed important to low-income Latino parents on their journey to healthier lifestyles, in the hopes of identifying beliefs that could be targeted by others designing interventions for this understudied group.

## 2. Methods

Focus groups were conducted with parents who participated in the Healthy Living Program (HeLP), or La Vida Saludable. HeLP is a community-based, family-inclusive childhood obesity treatment program serving low-income families in a predominantly Latino area of metropolitan Denver, Colorado, USA.

**Intervention Design**. While many weight management programs target at least part of the family as the unit of change, many typically only directly intervene upon a single child and a single caregiver [19]. The HeLP was designed to serve the entire family of a child with overweight or obesity, including parent, other primary caregivers and siblings who may be of any weight status. Primary care providers referred families to HeLP. These providers received four hours of training to use motivational interviewing to counsel families about childhood obesity and to refer families with at least one child above the 85th percentile for U.S. CDC BMI who reported interest in attending the intervention. To standardize the process for referral, families completed a tablet/computer-based lifestyle screening questionnaire, which generated a tailored counseling document that was reviewed by providers and families at all well-child visits [20]. This document depicted the child’s growth charts, offered suggestions for lifestyle change based on the family’s responses to questions about diet and activity habits, invited the family to consider attending HeLP if a child’s BMI exceeded the 85th percentile, and presented a pictorial and written description of the HeLP program. Providers were trained to refer those families expressing willingness to attend HeLP within the next few months, while leaving open the possibility for families facing time barriers to be referred at a future appointment.

The Healthy Living Program is designed to improve family functioning by building parenting skills and by directly engaging the entire family in planning, food shopping, and cooking family meals together as well as participating in physical activity together. The HeLP is also innovative in incorporating Share our Strength’s Cooking Matters for Families, hands-on nutrition, and a culinary curriculum designed to reduce food insecurity in low-income families [21]. Program content and delivery style were specifically tailored for cultural relevancy to a low-income Latino population. Some examples of tailoring delivery to this demographic group include the following: (1) family groups aimed at maximizing social support delivered by culturally concordant bilingual health educators; (2) culturally relevant foods in healthy cooking classes; (3) dance offered as a fitness option, which has been shown to increase physical activity in Latinas [22]; (4) written materials designed for participants with low literacy in English or Spanish; and (5) an overall program structure aimed at promoting the value of family togetherness with leveraging familism to drive behavior change [23]. Throughout the 12 educational and experiential sessions, the curriculum includes approximately 8 h of parenting skills training, 6 h of dietary management of excess weight, and 6 h of healthy nutrition appropriate for any weight status. The class format includes parent group discussions, family group discussions, hands-on cooking classes (parents and children), fitness classes for children, and a family meal offered during each session. For the first parenting discussion at the HeLP, a health educator reinforced the content of the providers’ MI counseling protocol by exploring parents’ attitudes about childhood obesity, its causes, its consequences, and potential reasons to act. Families set and track goals in the domains of parenting skills, nutrition, and fitness each week. The HeLP curriculum content has been described in further detail elsewhere and as a result of this curriculum, there was a significant mean reduction in relative BMI from pre- to post-intervention for children who were overweight and obese [8,18]. See Table 1 below for summary related to the content of the 12 educational and experiential sessions.

**Qualitative Study Design/Methods:** To complement our previously published quantitative study that demonstrated HeLP participants experienced a reduction in BMI, we completed focus groups to understand better how the intervention may have impactedbeliefs relevant to health behavior changes [8,18]. Parents who participated with their children in at least two of the twelve HeLP sessions were invited to participate in focus groups. Parents were contacted by telephone three to six months after completing the program and described the purpose and structure of the focus group. Parents who agreed to participate were offered several dates and selected the option most convenient to them. Reminder calls were conducted one day before each focus group. Focus groups were held approximately six months after families participated in the HeLP. Four focus groups were conducted in Spanish and two focus groups were conducted in English, a distribution that reflected the language preferences of the larger HeLP participant population. Each focus group lasted approximately 90 min and was conducted by a member of the research team trained in qualitative methods and fluent in both Spanish and English. The focus group moderator had completed formal post-graduate training in qualitative research methods and received ongoing mentoring from the qualitative methods expert on this research team. The focus group moderator utilized a semi-structured guide that was organized around constructs of the Theory of Planned Behavior. This theory posits that human behavior is guided by beliefs about the likely consequences or other attributes of a behavior (behavioral beliefs), the normative expectations of other people about a behavior (normative beliefs), and beliefs about the presence of factors that may further or hinder the performance of a behavior (control beliefs) [24,25]. The guide was designed to elicit parents’ experiences during the HeLP, to identify barriers to and facilitators of healthy lifestyle change, and to explore any possible changes in perceived beliefs attributed to the program. Specific domains of inquiry included parents’ experiences being referred to and participating in the HeLP, their experiences attempting the lifestyle changes discussed in the HeLP, their perceptions of how such changes fit within their daily lives, cultural traditions, and family customs, and how they felt other Latinos might perceive these changes. Faculty with expertise in qualitative methods and a HeLP program facilitator reviewed the moderator guide for theoretical coherence and clarity. The guide was consistently in all six focus groups.

Each focus group was digitally recorded and transcribed verbatim. Spanish recordings were transcribed in Spanish and then translated to English. Each focus group participant received a USD 30 gift card as compensation for their time, childcare was provided, and dinner was offered during each group. Focus groups were conducted until the research team ceased to find new themes, indicating that thematic saturation was achieved. The Institutional Review Board (blinded for review) approved this study.

**Data Analysis**. Data were analyzed using an iterative process involving established qualitative content methods and reflexive team analysis [26]. Focus group transcripts were independently analyzed multiple times by two members of the research team to achieve immersion. Coders were trained by the qualitative methodology expert in addition to having previous graduate coursework in coding methods. Code categories were developed independently using an emergent, rather than a priori, approach and then compared and discussed until code agreement was achieved [25,26,27,28]. Two members of the team applied the resulting codes to the transcripts, and the third confirmed the coding; consensus was achieved through reflexive team analysis. The study team met regularly to check new findings, discuss emergent new codes and themes, and assess the preliminary results of the analysis process [28]. The qualitative data software program ATLAS.ti v.7.0 (Scientific Software Development, GmbH, Berlin, Germany) was used for data organization and management.

## 3. Results

### 3.1. Participant Characteristics

Seventy-six families who had participated in the HeLP in the prior 6-month period were contacted to participate in this qualitative study. Of these, 37 parents were consented and enrolled in the to focus groups. Participants recruited attended a mean of 8 of 12 HeLP sessions. Details regarding participants’ demographic characteristics are presented in Table 2. The majority of participants > 50% were considered low income due to earnings < 175% percent of the Federal Poverty Level for a family of four at the time of the study.

### 3.2. Thematic Analysis

Several themes emerged in the analysis of the focus group data, including parents’ perspectives about barriers to and facilitators of a healthy lifestyle, which have been reported elsewhere [29]. The results below focus on how participants reported changes to their beliefs about (1) unhealthy weight, (2) nutrition, and (3) physical activity after participation in the HeLP.

#### 3.2.1. Category 1: Change in Beliefs Regarding Unhealth Weight

*Participant reflections on changes in beliefs*: *unhealthy weight*. Two themes emerged regarding parents’ reflections about changes in their beliefs about unhealthy weight in childhood that they attributed to participating in the intervention. These themes were: (1) decreased acceptance of unhealthy weight status in children, and (2) increased understanding of the importance of addressing weight early. See themes regarding changes in beliefs regarding weight and illustrative quotes presented in Table 3 below.

*Decreased acceptance of unhealthy weight status.* Parents indicated that, before they participated in the HeLP, they believed that the increased weight, causing an unhealthy BMI in their children, was not an issue and that neither parents nor children should worry about it. Many parents excused or rationalized their children’s weight. However, discussions with medical providers who were trained to use motivational interviewing to counsel about unhealthy weight at the time of referral, and group discussions during HeLP that reinforced the importance of taking action, may have influenced the shift to understanding that excess weight is not healthy for children. After participating in HeLP, the parents recognized specific implications for their health and their child’s health.

*Understanding of intervening on weight problems at an early age*. Parents discussed that, prior to their participation in the HeLP, they thought that their children were likely to outgrow being at an unhealthy weight without additional effort or attention to the matter. Participants noted that the counseling from medical providers that was reinforced at the HeLP prompted them to adopt the belief that addressing their children’s weight now had value for preventing future medical problems.

#### 3.2.2. Category 2: Change in Beliefs Regarding Nutrition

*Participant reflections on changes in Beliefs: nutrition.* Three themes emerged regarding how parents felt their beliefs about nutrition changed. (1) Parents’ beliefs around using food as a reward changed, (2) parents reported increased awareness of the benefits of a healthy diet, and (3) increased understanding about how to adapt traditional foods to fit a healthier diet. See Table 4 for themes regarding changes in beliefs around nutrition and illustrative quotes.

*Beliefs about using food as a reward.* Parents reflected that, before participating in the HeLP, it was normative to use food as a reward. However, parents reported that participation in the HeLP supported a transition to the belief that food should not be used as a reward, either for their children or for themselves. Parents became more aware of alternate methods of positive reinforcement.

*Increased awareness of the benefits of a healthy diet*. Following participation in the HeLP, parents reported that they appreciated a broader range of benefits of a healthy diet for children, contrasted with their lack of awareness of such benefits prior to the HeLP. Among the benefits that parents perceived following HeLP participation were that a healthy diet kept their children full, improved constipation, and helped to prevent disease.

*Awareness of adapting traditional foods to fit a healthy diet*. Participants described an evolution of their beliefs about the adaptability of traditional foods and eating habits. These participants recalled that, before the HeLP, even though some traditional foods and habits might not be particularly healthy, they were an immutable part of their culture. However, after participating in the HeLP, parents discussed a new acceptance that traditional foods and eating habits could be adapted to maximize healthfulness.

#### 3.2.3. Category 3: Change in Beliefs Regarding Physical Activity

*Participant reflections on changes in Beliefs: Physical Activity*. Two themes emerged regarding changes in participants’ thoughts about physical activity. (1) Parents reported changes in beliefs about access to physical activity and (2) increased awareness of the benefits of family exercise. See Table 5 for themes around changes in beliefs regarding physical therapy and illustrative quotes.

*Access to physical activity*. Participants reported that they previously believed that engaging in physical activity was prohibitively difficult, in part because being outdoors could be unsafe for their children. This fear of possible danger outdoors made many parents hesitant to encourage their children to engage in regular physical activity. Following participation in the HeLP, however, parents recognized several fun indoor alternatives to outside play. Participation in the HeLP also led parents to understand that the whole family could enjoy physical activity together.

*Benefits of physical activity*. Participants reported that, after their participation in the HeLP, they now have a new appreciation of the multiple benefits of physical activity for their children. Parents reported, to their surprise, a wide range of behavioral and quality-of-life benefits from the increased physical activity they observed. The benefits of physical activity discussed most by parents included that it helped their children sleep better, provided them with more energy during the day, and made them less moody.

## 4. Discussion

This study complements a prior quantitative analysis showing improvements in child BMI relative to the 95th percentile before and after the HeLP intervention [18]. Qualitative methodology using focus group was used for this study to allow participants to have reflexive conversations about what they felt was relevant for them in the program. Specifically, this study elucidates beliefs regarding child weight, nutrition, and physical activity that the HeLP changed in ways meaningful to low-income Latino parents.

To our knowledge, this is the first study to describe the transitions of beliefs reported by low-income Latino families attending a weight management program of any kind, particularly a program that is community-based and family-inclusive. The transitions in beliefs follow logically from the program structure and goals. The parents’ reflections suggest that the methods, including discussion, education, and facilitated practice of healthy lifestyle skills, synergistically contributed to evolved health beliefs. Specific aspects of HeLP design that influenced beliefs included coordinated messaging about child weight between referring providers and discussions at the program, hands-on cooking classes in which traditional Mexican foods were adapted, and structured goal setting with self-monitoring targeting health behaviors consistent with the changes in beliefs reported. The culturally tailored and family-inclusive setting that purposefully addressed socioeconomic barriers to health, including food insecurity, neighborhood safety for exercise, health literacy, and ethnic/language discrimination, may have increased the effectiveness of messaging in the HeLP setting over traditional health education messaging, which has often relied on personal/family responsibility to overcome such barriers [30].

In the previously published studies by the authors, families described aspects of the Healthy Living Program that helped parents to overcome cultural, economic, and behavioral barriers to healthy lifestyle changes, leading to improvements in child BMI pre- and post-intervention [8,18]. The current study takes a complementary look at the transitions in attitudes and beliefs that accompanied behavioral changes. Parents reported changes in beliefs that reflect the design and intent of the educational and skill-building activities provided in the intervention. Consistent with the Theory of Planned Behavior, program components worked to change subjective norms about child weight, beliefs about the benefits of healthy habits, and perceived control or self-efficacy that drives and sustains changes in health behaviors. The components of the HeLP that drove change might be generalized to other interventions for this population, including purposeful integration with clinical care, peer support, and heavy emphasis on experiential skill-building in a family-inclusive setting.

Previous research has documented that Latino parents are likely to perceive that an unhealthy degree of excess weight gain in a child is acceptable, a perception that reversed during HeLP [29,31]. The HeLP complemented advice about child weight from a medical provider trained in motivational interviewing with peer discussions led by culturally concordant health educators. Such a combination of components may be helpful in future programs that focus on this transition in beliefs about excess weight.

The reported changes in beliefs about nutrition also follow logically from the structure and content of the Healthy Living Program. Parents reported a transition away from the belief in food as a reward toward alternative means of positive reinforcement. Studies show that the use of food as a reward further increases the craving for highly palatable energy-dense foods [32]. The HeLP nutrition classes openly explored culturally engrained and inter-generationally transmitted eating practices. This discussion was followed by parenting classes that trained parents in alternative methods of positive reinforcement and in skills to redirect children’s requests for preferred, but unhealthy, foods. A second transition noted via self-reflection by participants regarding nutrition-related beliefs was a new awareness of the benefits of a nutritious diet, which was interwoven throughout the nutrition curriculum and reinforced through weekly goal-tracking forms. The third nutrition theme was the transition from cultural determinism of cooking and eating practices to belief in easy adaptation for improved healthfulness, while maintaining acceptance by the family. Extensive literature has documented the need for hands-on learning when it comes to cooking healthier versions of traditional foods [33,34]. With this in mind, HeLP included six hands-on cooking sessions that allowed families to prepare and consume healthier versions of traditional cuisine. Thus, the repeated exposure to healthy substitutions, including hands-on practice, seems important to alter parents’ normative beliefs and improve self-efficacy for making healthy substitutions to traditional cuisine. Additionally, qualitative studies have noted that Latino parents accepted healthy substitutes into traditional Latino meals and found them to still be palatable for parents and children [33,34].

Changes to beliefs about physical activity similarly followed from the structure and content of the HeLP. Participation in the HeLP changed normative beliefs about safety and accessibility of physical activity and control beliefs about motivation to participate in physical activity. These changes followed exposure to indoor and outdoor play options, and discussions about the multiple benefits of physical activity reinforced by weekly physical activity goal setting. Parents reported seeing improvements in their children’s sleep and mood, reinforcing their new belief regarding the benefits of PA. HeLP targeted all family members that were active in the child’s care. This was intentional, as prior research has found that, to make sustainable changes in lifestyle behaviors related to diet and physical activity, all key family members need to be involved [35]. Being active as a family became rewarding and allowed parents more quality time with their children. As having multiple generations in one home is common in Latino families, the involvement of the entire family in physical activity led to sustained changes in beliefs and habits.

This study provides important guidance for the design and evaluation of weight management programs serving low-income, Latino populations. This community-based childhood obesity intervention is unique in that it was designed to ensure a family-inclusive approach. For Latino parents who place a strong emphasis on the value of family, or Familism, this approach helped to solidify change for the entire family and not just the child who had an unhealthy weight [36]. This study may inform the development or refinement of weight management programs seeking to serve this population. The assessment of changes in the beliefs reported in this paper might be valuable as a measure supporting quantitative program outcomes.

The limitations of this study, common to qualitative analyses, include the inherent subjectivity and possible transmission of preconceptions of the study team. However, we took thoughtful measures to reduce subjectivity, including engaging in rigorous reflexive team analysis and emphasizing the principles of grounded theory. Further, the members of the study team involved in the thematic analysis did not participate in the design of the HeLP. The responses of parents about past beliefs are retrospective reports, as focus groups were conducted after the intervention was completed. Most families participating in HeLP attended more than half of the sessions, so we are limited to presenting results from focus groups of participants who completed the program, as there were few dropouts. The overwhelming majority of participants reported Mexican ancestry, so generalization to other Latino cultural groups is limited.

## 5. Conclusions

In conclusion, the inclusion of the entire family in a weight management intervention targeted to low-income Latino communities appears to be an important aspect of the intervention’s success in modifying caregiver beliefs. Through using qualitative research methods, our findings will help future researchers to understand better how perspectives may change beliefs in this community and that there is evidence that this change may be necessary for improvements in youth health behaviors. Future studies are needed to complete a more in-depth analysis of the change in beliefs by the use of interviews or focus groups prior to the intervention, and then post-intervention. This would allow a better understanding of how a change in beliefs allowed for or assisted in a behavior change, and whether these changes in beliefs will help in long-term outcomes of healthier weight/BMI due to sustained change in lifestyle behaviors.

## Figures and Tables

**Table 1 behavsci-12-00073-t001:** Summary of Curriculum Content.

Parent Support	Nutrition	Physical Activity	Teaching Methods	Skills Employed
Build healthy home environment ▪ Family meals ▪ Rules and responsibilities at mealtime ▪ Gain support from family and friends Understand reasons for weight gain Overcome Food Neophobia Parenting skills: ▪ Praising, ignoring, timeout, time in, chore-grounding ▪ Giving effective directions ▪ Parenting as a team Plan ahead for changes Foster a healthy body image	Involve children in food preparation ▪ fruits and vegetables at every meal ▪ Shopping with kids My Plate Read nutrition labels Cook with: whole grains, healthier fats, produce, lean proteins Healthy breakfasts, healthy snacks Add produce to recipes Healthy snacks Recipe outlines Time saving techniques Make healthier restaurant choices	Reduce Screen time Identify barriers & solutions Stay motivated Family fitness Overcome burnout Plan for flexibility	Dialogue with group Self-reflection Information Skill training Role play Modeling Skill building with guided practice Problem solving Social support Application Sampling new food Grocery store tour Menu planning activities	Self-monitoring Goal Setting ▪ Specific ▪ Measurable ▪ Accountability ▪ Rewards ▪ Realistic ▪ Time limited Self-efficacy Skill Practice Family discussions Parent as role model

**Table 2 behavsci-12-00073-t002:** Focus group participant demographic characteristics.

Variable	Participants n = 37 (% of Respondents)
Gender	
Women	35 (95%)
Ethnicity	
Hispanic	28 (76%)
Parents of a pre-school-aged child	20 (58%)
Parents of a male overweight child	18 (55%)
Parents of a female overweight child	15 (45%)
Parents with 1 minor child	3 (9%)
Parents with 2 minor children	12 (35%)
Parents with 3 minor children	11 (32%)
Parents with 4 or more minor children	8 (24%)
Missing data of children not reported	3 (8%)
1 adult at home	4 (15%)
2 adults at home	20 (74%)
3 adults at home	3 (11%)
Missing Data of number of parents not reported	10 (27%)
Income Based on family of 4	
<USD 1900/month (Approx. < 100% of FPL-2015)	18 (49%)
USD 1900–3500/month (approx. 100–175% of FPL-2015)	7 (19%)
>USD 3500/month (Approx. > 175% of FPL-2015)	2 (5%)
Maternal Educational Attainment	
Did not complete high school	12 (43%)
High school diploma	7 (25%)
Some college attendance/completed college	9 (32%)
Missing data	9 (24%)

**Table 3 behavsci-12-00073-t003:** Changes in beliefs regarding unhealthy weight and illustrative quotes.

Theme	Illustrative Quotes: Reflections on Prior Beliefs	Illustrative Quotes: Reflections on Current Beliefs
Decreased acceptance of unhealthy weight status	“My kid ain’t fat. She’s big-boned! You know?”	“I’d better do something [about my child’s weight]. You know, because unhealthy could mean dangerous. Unhealthy could mean more asthma.”
Increased understanding of the importance of addressing weight at an early age and consequences of unhealthy weight	(1) “They are young. They are growing. The [extra] weight, you know, they’ll grow out of it.’” (2) “That’s baby fat… it’s just baby fat. [My daughter will] lose it, you know.”	(1) “I [am now] concerned about diabetes. My dad had type 2. I’ve gained 70 pounds in the past 2 years. I don’t want to see us go down that road.” “I liked it that [the nurse practitioner] told me the truth, so we can get [my daughter’s eating habits] better. Because diabetes runs in my family and so I want to keep that under control, so she doesn’t have to take medicine or anything later.”

**Table 4 behavsci-12-00073-t004:** Changes in beliefs regarding nutrition.

Themes	Illustrative Quotes: Reflections on Prior Beliefs	Illustrative Quotes: Reflecting on Current Beliefs
Beliefs about using food as a reward	(1) “It’s really hard, even as a grown-up, as their mom [not to use food as a reward]. ‘Yeah, an ice cream sounds great. You got a 4.0? Let’s go get some ice cream!’ You know, it’s real easy to do, reward with food.” (2) “My kids were a little fat because I used to work a lot, and I thought that [a way of] compensating the time that I was not with them was [to] buy food, like junk food.” (3) “Before the Healthy Living Program, I didn’t think I had a problem. I just thought I like to eat. I liked food and I liked to reward people with food.”	(1) “My mom was a big food rewarder. And I’m trying not to do that. I’m trying to find other things to use as rewards other than food. ‘Cause, my mom was such a big food rewarder and in the Healthy Living [Program] classes, you know, looking at my behavior, it was like ‘Uh oh, I need to pull back the reins on that’.” (2) “After Healthy Living, I found out there are other things other than food that give me pleasure.”
Increased awareness of other benefits of a healthy diet		(1) “We all [in the family] get full and do not want anything else. If you buy [vegetables], that [is what] happens with my family…they don’t even have cravings [for junk food] with all the salad they ate!” (2) “[Fruits] are good to fight constipation. Our system works better [when we eat fruits].” (3) “[Fruits and vegetables] are a natural source of vitamins. And it helps your immune system… [my children] get sick less.”
Awareness of adapting traditional foods to fit a healthy diet	(1) “We struggled with starch too [prior to HeLP]. Probably like a culture thing. Like, ‘You have to have tortillas to eat that. You have to!’ It is like, ‘How do you eat that without a tortilla?’” (2) “Our customs over there [in Mexico], we do not eat portions; over there our plate is sometimes chili with meat and beans, pure protein and sometimes without vegetables.”	(1) “We have to change the habits we bring from [Mexico]. Like right now… we are learning about how to use the oven, or [to substitute] the turkey like that. I was the one that used to cook the beans with lard, to fry them, and now I do not have cravings for [that].” (2) “We heard [in the program], we changed the bigger plates to the smaller ones because our customs were to not get up until you finish your plate.”

**Table 5 behavsci-12-00073-t005:** Changes in beliefs regarding physical activity.

Themes	Illustrative Quotes: Reflections on Prior Beliefs	Illustrative Quotes: Reflecting on Current Beliefs
Access to physical activity: unsafe environment	“When I was younger, I was always outside. But now, the way people are, I have to be outside with [my children].”	“[My children] have the X-Box and they play a lot, those games to jump, to dance…. because right now it is [too] cold to go out and it has been a good entertainment for them this season.” (2) “The children feel more motivated with mom and dad, so the whole family is together. [When] we all go for a walk, swimming, or doing a sport, they feel more motivated.”
Benefits of physical activity		(1) “[My children] also sleep better if they get a lot of exercise.” (2) “[With physical activity], I see [my children] more alert, more focused… with more energy.” (3) “We all get along better [when we engage in physical activity]. [My children] are laughing more… they are not so cranky and fighting.”

## Data Availability

The data presented in this study are available on request from the corresponding author. The data are not publicly available due to sensitive nature of potentially identifiable information contained within full transcripts of the focus groups.

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
