# Peer review of "Exploration of Changes in Low-Income Latino Families’ Beliefs about Obesity, Nutrition, and Physical Activity: A Qualitative Post-Intervention Study"

_behavsci, 2022, doi:10.3390/bs12030073_

Round 1

Reviewer 1 Report

Behavsci-1537784 Changes in Low-Income Latino Families’ Beliefs about Obesity, Nutrition, and Physical Activity: A Qualitative Study Post-Intervention

This submission describes reflective results of participants that completed the Health Living Program (HeLP) also known as La Vida Saludable. The six-month program is aimed at educating and improving family wellness through the engagement of entire families through twelve experiential and educational experiences. Of the seventy-six families that completed the program, thirty-seven parents representing thirty-three of the completing families, were recruited to participate in post-program focus groups. This submission describes and assesses the results of these focus groups.

As presented, this is a rather lengthy discussion summarizing the focus group results. The authors indicate three overall areas of change:  obesity, nutrition, and belief (regarding activity). For each overall area, the authors indicate three or two primary themes. For obesity, decreased acceptance that overweight means healthy, and increased interest in addressing overweight/obesity among children. For Nutrition, the three themes are identified as reducing the use of food as a reward, understanding components of a healthy diet, and adapting traditional foods into healthier options. For the Beliefs (activity) area, the two themes identified were  changing attitude about being physically active and enhancing physical activity as a family.

Considering the outcomes being presented, this manuscript could have a much greater impact with a substantial revision. The introduction should be more focused, first presenting the problem (directly and succinctly) along with the theory importance of changing beliefs (behavioral theory) for effectiveness of interventions. The methods then should briefly describe and cite the La Vida Saluable study. Sufficient details would indicate the main aims of the twelve study experiences and then the specific details of the current focus-group based assessment.

Results should include a brief table of the characteristics of the focus group study participants (currently this is supplementary information- this table could be formatted, more clearly) followed by a second table summarizing the three main areas of content agreement among focus group participants, and the primary themes included in each area. The authors could include on primary example of parents statements that summarize each themes in a additional column in the results table but these example statements should be limited and not form the bulk of the manuscript submission. The discussion would then summarize the emergent post-study areas and themes and their link to the content of the experiential components of La Vida Saludable Study.

The authors are presenting information that could be useful for others to consider when designing studies. Demonstrating changes in belief systems are critical for behavior change and this focus group study appears to indicate that the HeLP/ La Vida Saludable effectively changed caregiver beliefs. However in the current, overly descriptive submission, the findings and potential impact of these findings are being lost.  A focused approach will strengthen this manuscript’s contribution to the literature.

Author Response

Please see revised manuscript attached 

Response to Reviewer 1 Comments

Thank you for taking the time to review this current revised manuscript. Below you will find the points noted by the reviewer and our response to each of the suggested comments.

Point 1: The introduction should be more focused, first presenting the problem (directly and succinctly).

Response 1: Thank you for the input regarding the introduction, to be more focused and succinct, the original 1st paragraph was deleted. The first paragraph now directly discusses the problem and its significance. Please see 1st paragraph lines 39-53.

Point 2: …along with the theory importance of changing beliefs (behavioral theory) for effectiveness of interventions

Response 2: In response to the recommendation to discuss the importance of the theory there is added explanation of the importance of the use of theory for the qualitative study that was done and why the behavioral theory was used. Please see lines 80-85, 102-110. Additionally in the methods section lines 180-188. For the larger intervention, the Theory of Planned Behavior change was not the guiding theory.  However, the overarching model of behavior change for HeLP and the PCP training for referrals was the Transtheoretical Model of Behavior Change. This broader model for behavior change includes the elements of the TPB

Point 3: The methods then should briefly describe and cite the La Vida Saludable study. Sufficient details would indicate the main aims of the twelve study experiences and then the specific details of the current focus-group-based assessment.

Response 3:

  • To respond to this reviewer comment, a table was added that shows a summary of the 12 aims of the educational and experiential sessions. See page 4 in the manuscript.
  • A separate heading was created to highlight the qualitative study design vs the Intervention design in the methods section (lines 116 and 164) so that it was clear the difference in the design of the larger interventions. While we appreciate the need to be succinct and the suggestion made to help in the organization of the methods section, we decided that additional background information regarding the larger intervention was needed as context for the discussion.
  • There is now an additional header for the methods section of the qualitative design/methods, see line 164.
  • There is also additional information on how the Theory of Planned Behavior was used in the qualitative assessment and why that theory was chosen in the methods section lines 180-188.
  • Also, an additional heading was created to clarify the data analysis that was done for the qualitative study.
  • As several changes were made, please see highlighted areas of the Methods section for these changes.

Point 4: Results should include a brief table of the characteristics of the focus group study participants (currently this is supplementary information- this table could be formatted, more clearly)

Response 4: The table for participant demographics was integrated into the text and made headings clearer with changes to show % of respondents. See Table 2 on page 6.

Point 5: followed by a second table summarizing the three main areas of content agreement among focus group participants, and the primary themes included in each area. The authors could include one primary example of parents’ statements that summarize each theme in an additional column in the results table but these example statements should be limited and not form the bulk of the manuscript submission.

Response to Point 5:

We appreciate the suggestion regarding re-organization of the results, and we have created tables throughout the text for clarity with each of the themes for ease of reading as well as highlighting when appropriate the contrast of pre-and post-reflections while maintaining the quotes in the contexts of the summary that is in the text.  This was noted to be done in another qualitative paper that was published in the special issue section so we felt it would be appropriate to use this format as well. Quotes were adjusted accordingly to ensure that contrast was clear from the perspective of the participant parents.

Point 6: The discussion would then summarize the emergent post-study areas and themes and their link to the content of the experiential components of La Vida Saludable Study.

Response to Point 6: Thank you for the suggestions regarding the content of the discussion. We looked at the discussion and decided it needed to be re-written for clarity, please see the discussion section in its entirety. Because there are extensive changes all track changes were not kept for ease of reading for reviewers.

Reviewer 2 Report

The article requires organization in its structure: It is considered that the conclusions continue to develop content derived from the results and continue to specify questions of methodology. This can make it difficult to understand this section. Obviously, this does not imply that the results and implications of the methodology are not referred to in relation to the conclusions of the study.

Regarding the objectives, it is indicated “The objective of this study was to examine the reflections after the interaction of the parents on how and why their beliefs about childhood obesity, diet and physical activity changed through the lens of the theory of planned behavior ”. In this objective, the result to be investigated is taken for granted. That is, it is assumed that there was a change in belief, as the title of the article announces. It would seem then that there was already an evaluation prior to this study, now corresponding to evaluate how and why beliefs changed. If so, it is appropriate to make it explicit. Otherwise, it is convenient to clarify and make the objective statements in different parts of the article coherent.

It would give methodological precision to the results. It would be convenient to describe the characteristics of the subjects participating in the qualitative study, taking into account the number of subjects participating in the Healthy Living program.

 The relevance of the characteristics of the subjects: low income is not explained in the document. Reference is made to the cultural link between origin and obesity only.

It would be relevant to present the focus group guide used since it constitutes the only evaluation instrument of this study.

It would be opportune to justify why Qualitative Methodology is chosen for this evaluation of the Healthy Living program.

Author Response

Please see revised manuscript attached 

Responses to Reviewer 2

Thank you for taking the time to review this current revised manuscript. Below you will find the points noted by the reviewer and our response to each of the suggested comments.

Point 1: The article requires organization in its structure

Response to Point 1:

Thank you for the detailed comments that have allowed us to re-organize the content of the manuscript for better flow and understanding.

Point 2: It is considered that the conclusions continue to develop content derived from the results and continue to specify questions of methodology. This can make it difficult to understand this section

Response to Point 2:  We appreciate the author pointing out the additional information that was noted in the conclusion. The reason for this was an omission on our part. This section should have been the discussion section. We have now labeled this correctly. We have also corrected this by putting the appropriate headers in place and there is now a distinct difference between the discussion portion and the conclusions. There have been extensive changes to the discussion section. So please see it in its entirety. Starting on line 438.

Point 3: In this objective, the result to be investigated is taken for granted. That is, it is assumed that there was a change in belief, as the title of the article announces. It would seem then that there was already an evaluation prior to this study, now corresponding to evaluate how and why beliefs changed. If so, it is appropriate to make it explicit. Otherwise, it is convenient to clarify and make the objective statements in different parts of the article coherent.

Response to Point 3: We have changed the title to reflect the changes in beliefs to indicate that it was not assumed but explored as a part of the analysis.  Thank you for bringing this to our attention, we have made changes on lines 134-143 to clarify that there was not an evaluation done prior and the qualitative study was an evaluation of the intervention, done to improve the intervention as well as to see if there were any changes in beliefs through the course of the intervention, that helped lead to sustained behavior changes.

Point 4: It would be convenient to describe the characteristics of the subjects participating in the qualitative study, taking into account the number of subjects participating in the Healthy Living program.

Response to Point 4: Please see the table2 of participant characteristics, which is now in the manuscript itself for ease of reading.  Additionally, see lines 249-252 which states the number of participants in the larger Healthy Living Program at the time the qualitative study was completed.

Point 5: The relevance of the characteristics of the subjects: low income is not explained in the document.

Response to Point 5: Thank you for pointing out this omission in explaining how we defined low income. To correct this Table 2 now has additional text that explains that the US Federal Poverty level for a family of 4 monthly income was used. Additionally, the text has been added to results to explain the definition of low-income. See lines 254-255.

Point 6: It would be relevant to present the focus group guide used since it constitutes the only evaluation instrument of this study.

Response to point 6: This focus group guide will be added as supplementary material for reviewers to look at questions that were asked in the focus groups for this qualitative study.

Point 7: It would be opportune to justify why Qualitative Methodology is chosen for this evaluation of the Healthy Living program.

Response to Point 7: Please see lines 442-444 of the discussion which clarifies why qualitative methodology was used for this study.

Reviewer 3 Report

The work is unique and I enjoyed reading but please check the comments in the attached manuscript, they can help to improve the work.

Author Response

Please see the revised manuscript which is attached 

Response to Reviewer 3 Comments

The points noted in the text have been summarized as I was unable to cut and paste directly from the sticky notes into a document to respond to the reviewer’s comments. We appreciate the detail that the reviewer suggested to make this current manuscript clearer and more concise. Thank you for taking the time to review this current revised manuscript.  Below you will find the points noted by the reviewer and our response to each of the suggested comments.

Point 1:

Introduction comments stated similarly to Reviewer 1 that the introduction should start with the background and the significance as well as specific hypotheses.

Response to Point 1:  This has been improved by deleting the first paragraph which now makes it more concise and clearer what the main objective and the significance of the study population is.  Please see the re-written Introduction section starting on page 1. The objective and hypotheses (or lack of) are specifically noted in lines 103-110 are highlighted for review.

Point 2: Citation of sources. Thank you for the comment regarding ensuring that citations are noted for works that have been consulted.

Response to Point 2: A review of the citations was done during revisions of the manuscript and citations of all sources are noted in the document which corresponds to the bibliography.

Point 3: To have a separate literature review section

Response to Point 3:

Thank you for your suggestion of a literature review section but we have decided the current formatting which is noted in the instructions for authors has been followed and citations are noted for both empirical literature and theoretical literature throughout the paper in the appropriate sections.

Point 4: If paper went through ethical clearance

Response to Point 4: The Institutional review board approved the study which is the ethical clearance you are referring to and this information with IRB #, was sent to the Editors of the journal as well.

Point 5: No statistical participant section.

Response to Point 5:  In Table 2 it now shows the descriptive statistics and is included directly in the manuscript. Please see table 2 on page 6 of the revised manuscript

Point 6: Use of Tables to summarize finding

Response to Point 6: Please see the Results section in its entirety which was revised to have additional tables by themes interdigitated throughout the section for ease of reading and clarity. The summary of these findings is now more succinct in the body of the results section of the manuscript. This format was noted in a published paper in the Special Issue portion of the journal so we felt it would be appropriate to use as well.

Point 7: Conclusion is too long- state purpose of the study, the methods used and results

Response to Point 7: There was an omission on our part which was brought to our attention by another reviewer, in that the conclusion section should have been the discussion section. The discussion section has now been significantly revised. Please see the discussion portion starting on page 10 where the suggested components have been added.

Round 2

Reviewer 3 Report

Authors have addressed all the issues